# Evaluation of Salivary Biomarkers of Periodontal Disease Based on Smoking Status: A Systematic Review

**DOI:** 10.3390/ijerph192114619

**Published:** 2022-11-07

**Authors:** Jin-won Noh, Jong-Hwa Jang, Hae-Soo Yoon, Kyoung-Beom Kim, Min-Hee Heo, Ha-eun Jang, Young-Jin Kim, Yejin Lee

**Affiliations:** 1Division of Health Administration, College of Software and Digital Healthcare Convergence, Yonsei University, Gangwon-do, Wonju 26493, Korea; 2Department of Dental Hygiene, College of Health Science, Dankook University, Chungcheongnam-do, Cheonan 31116, Korea; 3Department of Public Health Science, Dankook University Graduate School, Chungcheongnam-do, Cheonan 31116, Korea; 4Department of Health Administration, Dankook University, Chungcheongnam-do, Cheonan 31116, Korea; 5Industry-Academic Cooperation Foundation, Yonsei University, Gangwon-do, Wonju 26493, Korea; 6Department of Health Administration, Yonsei University Graduate School, Gangwon-do, Wonju 26493, Korea; 7Department of Healthcare Management, College of Bio Convergence, Eulji University, Gyeonggi-do, Seongnam 13135, Korea; 8Department of Public Health Graduate School, Korea University, Seoul 02841, Korea

**Keywords:** biomarker, diagnosis, periodontal diseases, review, saliva, smoking

## Abstract

Saliva is a useful biomarker for diagnosing oral health conditions, including periodontal disease (PD). Smoking is a risk factor for PD. The aim of this systematic review was to summarize the salivary biomarkers associated with PD based on smoking status. A comprehensive search of the MEDLINE (via PubMed), EMBASE, Cochrane, SCOPUS, and Web of Sciences databases was conducted up to 1 January 2021 using key terms relevant to the topic of our research and Cochrane methodology and improved with searching a gray literature resource. The methodological quality of all included studies was assessed with the revised Quality Assessment of Diagnostic Accuracy Studies-2. Seven studies were included. Smokers had increased levels of malondialdehyde, sialic acid, salivary cortisol, salivary interleukin 1β, albumin, tissue inhibitor of matrix metalloproteinase (TIMP), and the pyridinoline cross-linked carboxyterminal telopeptide of type I collagen (ICTP), as well as decreased levels of superoxide dismutase, activity of lactate dehydrogenase, activity of enzyme activity of β-glucuronidase, uric acid, matrix metalloproteinase-8 (MMP-8)/TIMP-1 ratio, and combinations of MMP-8 and ICTP. However, mixed results were observed some studies in detecting glutathione peroxidase, MMP-8, and MMP-14. The results were interpreted with caution because of limitations in the number of included studies and the study design. Some salivary biomarkers are potentially useful in combination or alone for diagnosing PD. Methodological and systematic studies are needed to develop more effective biomarkers.

## 1. Introduction

Periodontal disease (PD) is one of the most common inflammatory diseases of the oral cavity, and it affects up to 90% of the global population [1]. It is caused by inflammation of the surrounding structures of teeth, such as the gingiva, periodontal ligament, and bone; if not treated properly, it can lead to tooth loss and contribute to systemic inflammation [2].

Since PD often progresses without symptoms, many patients do not receive professional dental care until the periodontal destruction that cannot be treated has progressed [3]. In addition, there is an unmet need for diagnosing PD quickly because a PD diagnosis relies on time-consuming clinical measurements [3]. Saliva is an optimal biological fluid to serve as a point-of-care (POC) diagnostic tool for PD. From this point of view, a POC diagnosis simplifies diagnosis and improves prognosis, and the feasibility of PD diagnostic testing has been reported [4]. 

Many promising salivary biomarkers associated with PD have been reported [3]. The pathogenesis of periodontitis is related to enzymatic alterations such as malondialdehyde (MDA), sialic acid (SA), lactate dehydrogenase (LDH), cortisol, β-glucuronidase (BetaG), interleukin 1β (IL-1β), antioxidants, oxidative stress, superoxide dismutase (SOD), 8-hydroxydeoxyguanosine, glutathione peroxidase (GPx), and 4-hydroxynonenal [5,6,7,8]. SOD is an antioxidant enzyme that is localized within human periodontal ligaments, and it provides an important defense within gingival fibroblasts against superoxide [9]. However, plasma glutathione peroxidase, a selenium-containing peroxidase, comprises a major group of enzymes that remove the hydrogen peroxide created by SOD in the cell [10]. IL-1β stimulates the expression of matrix metalloproteinases (MMPs), which contribute to bone resorption and tissue destruction [11]. To date, 24 different MMPs have been cloned, and three of them have been found in humans. Based on the substrate to be degraded, they are divided into six types: collagenase, gelatinases (type collagenase), stromelysins, matrilysins, membrane-type metalloproteinases, and others [12]. Among the MMPs, MMP-8 and MMP-9 are in the spotlight as biomarkers for periodontal disease. A kit that can test for MMP-8 in 5 min in an office has been developed [13,14].

PD progression can be influenced by various risk factors such as periodontal pathogens, host factors, anatomical factors, and iatrogenic factors [15]. Among the associated risk factors, smoking is the second-largest risk factor for PD after dental plaque [1]. Reports indicate that the prevalence of periodontitis is 3–6 times higher in smokers than in non-smokers, and the increased risk is proportional to the duration of smoking and smoking rate [16,17]. Smokers exhibit more pronounced PD clinical findings than non-smokers, such as deeper pockets, more extensive and severe loss of attachment, higher levels of bone destruction, and higher rates of tooth loss [18,19,20]. In addition, smoking negatively affects successful implant placement and non-surgical and surgical treatment [21].

Meanwhile, saliva contains a unique and complex variety of enzymes and proteins with important oral functions [5]. The use of these enzymes for diagnosing PD has unfortunately been hindered because the relevance of protein and enzymes in saliva and disease etiology remain limited. Furthermore, enzymatic alterations can be caused by various factors such as temperature, pH, enzyme substrates, and the effect of inhibitors and activators [22]. In particular, tobacco compounds the damage activities of salivary enzymes at the molecular level [23]. However, saliva samples are non-invasive, readily available, and inexpensive; therefore, saliva can be a valid alternative to blood as a biomarker [24,25]. Saliva is a favorable oral fluid to determine the health state of the oral cavity, including the presence of PD [26,27]. Therefore, an effective and reproducible salivary biomarker would be preferred over other biomarkers. The aim of this study was to evaluate the evidence, using a systematic review, and to highlight the future directions regarding the diagnostic potential of salivary biomarkers associated with PD based on smoking status.

## 2. Materials and Methods

### 2.1. Study Design

This study was a systematic literature review that synthesized 20 years of research by analyzing PD-related salivary biomarker factors associated with smoking status. We conducted this systematic review per the standard method of the Preferred Reporting Items for Systematic Reviews and Meta-Analyses (PRISMA) guidelines [28] and according to the PRISMA 2020 checklist (Appendix A) [29].

### 2.2. Literature Search and Selection

The review questions were formulated in the population, intervention, comparison, outcome, and type of study design (PICOT) format [30] (Table 1). The search strategy consisted of three main concepts: target condition (PD), type of oral sample analyzed (saliva), and index tests (salivary biomarkers). Plain text words (including synonyms or plural forms) and controlled vocabulary of concept (e.g., Medical Subject Headings terms) were combined and used for searches in the title and abstract fields for each database.

The electronic database search was conducted in the MEDLINE (via PubMed), EMBASE, SCOPUS, and Web of Sciences databases. The search was restricted by the following specification: English-language literature published in peer-reviewed journals from January 2000 to January 2021. The complete search strategy is available in Appendix A. The target conditions were PD, based on the 2017 classification of periodontal and peri-implant diseases and conditions published by the American Academy of Periodontology (AAP) and the European Federation of Periodontology (EFP), irrespective of the severity or extent of the illness [31,32].

The detailed selection process is presented in Figure 1. The initial database search returned 3702 articles; 2056 duplicates were removed. Among the remaining 1646 articles, 1426 articles were excluded in the abstract evaluation phase. A full-text evaluation was conducted for the 220 retained articles; 213 articles were excluded, based on the criteria for selection (e.g., the articles did not report smoking status-specific values). Finally, seven articles were included for the systematic review.

The seven studies used for the data analysis were coded based on the year of publication, country, study design, case sample size, age, definition of PD, smoking status, and type of salivary biomarkers for the subgroup analysis on the major categorical variables. After categorizing these factors, the related factors of the group and main conclusions were coded. For effect size, the sign, value, and correlation effect size were coded. 

### 2.3. Study Selection and Exclusion Criteria

We included studies if (1) they included individuals with clinical PD but without explicit systemic disease; (2) they provided individual level, smoking status-specific results of at least 20 individuals; (3) they were prospective studies; and (4) they were written in English and published in a peer-reviewed journal.

We excluded studies if any of the following criteria were present: (1) they explicitly stated that they included individuals with systemic disease, conditions, or syndromes; (2) they were retrospective, prognostic, or predictive accuracy studies; (3) they were published in non-English language literature; (4) they involved non-human subjects; (5) they reported the data of fewer than 20 individuals for one smoking status; (6) they were published in theses, dissertations, reviews, letters, personal opinions, book chapters, short communications, conference abstracts, and patents; and (7) they did not report the smoking status-specific results of salivary biomarkers. In cases of overlapping articles of the same study cohorts, only articles with the largest number of participants were included.

## 3. Results

The systematic literature review yielded a total of seven papers for analysis. Of these, 213 articles were excluded because smoking status was not reported [9,13,33] or because they entailed a topic unrelated to the study question [18,34]. 

### 3.1. Risk of Bias Assessment

The quality assessment of all the included studies was independently reviewed by all authors by using a modified version of the Quality Assessment of Diagnostic Accuracy Studies (QUADAS-2) tool [35]. Salivary samples can be preserved for several years with little damage and degradation of the salivary molecules [36,37,38]; therefore, the time lag between a reference standard (i.e., clinical measurements) and the index test (e.g., saliva analysis) should not be a high-risk bias. However, standardized storage protocol must be followed completely to conserve the stability of the salivary biomarkers [38,39]. Therefore, we additionally included information on salivary sample storage in the modified version of the QUADAS-2 tool for the assessment of the bias.

In total, 14 salivary protein biomarkers were identified. Each case was described by measurements of the PPD and CAL; however, diverse thresholds and clinical parameters were used for diagnosing PD. The results of our quality assessment with the modified QUADAS-2 tools are summarized in Table 2. Except for two studies [5,40], all seven studies recruited study participants via non-random, convenience-sampling methods. Therefore, the risk of patient selection bias was marked as “high” for five studies. The risk bias of index test was recorded as “unclear” for all seven studies because the blinding assessments for measuring reference tests of interpreting the index test were not mentioned.

### 3.2. Descriptive Summary of the Studies Included in the Systematic Review

A descriptive summary of the studies included in this systematic review study is presented in Table 3. Most of the selected studies had been published in the previous 10 years and were conducted in three countries: India [5,6,7,40,41], Turkey [8], and Finland [42]. The number of participants with PD ranged from 40 [41] to 100 [5]. Fourteen different types of salivary biomarkers were evaluated. One study [42] reported salivary biomarkers, stand-alone, and combination/ratio, whereas other studies reported the results for only single biomarkers.

### 3.3. Salivary Biomarker Levels Based on Smoking Status

Table 4 summarizes salivary biomarker levels based on smoking status. Compared to non-smokers, smokers had increased levels of MDA [6], SA [6], salivary cortisol, and IL-1*β* [7] (*p* < 0.001). The tissue inhibitor of matrix metalloproteinase (TIMP) [42] and pyridinoline cross-linked carboxyterminal telopeptide of type I collagen (ICTP) [42] were higher in smokers than in non-smokers but without statistical significance. In addition, albumin [40] was higher in smokers than in non-smokers, but this was not statistically significant (*p* > 0.05). 

On the other hand, the levels of SOD [6] (*p* < 0.001) and uric acid (UA) [40] (*p* < 0.01) using spectrophotometry were significantly higher in non-smokers than in smokers. The levels of activity of LDH [5] and BetaG [5], uric acid (UA) [40], MMP-8/TIMP-1 ratio [42], and combination of MMP-8 and ICTP [42] were higher in non-smokers than in smokers but without statistical significance.

In particular, the salivary biomarkers GPx [6,8] and MMP-8 [41,42] showed conflicting results between smokers and non-smokers when different detection methods were used. These results suggest that further studies are necessary. In addition, the descriptive statistics varied among the studies on MMP-8 using enzyme-linked immunosorbent assay (ELISA), where the level of MMP-14 was higher with APMA than without APMA.

## 4. Discussion

This systematic review estimated the salivary biomarkers of PD based on smoking status. In our results, seven studies published since 2000 highlighted 14 host-derived salivary biomarkers. Owing to the considerable and growing interest in saliva as a useful tool for biomarker analysis, as highlighted based on our literature search, significant studies have been recently published [3,4,38]. Salivary-derived diagnostic techniques could potentially allow for the timely screening of an entire population for specific diseases [3,33]. However, studies evaluating the effectiveness of salivary biomarkers for diagnosing PD are in their infancy [38].

### 4.1. Evidence for Salivary Biomarkers Based on Smoking Status

As a result of a systematic literature review on salivary biomarkers for PD diagnosis by smoking status, some markers showed significant differences between smokers and non-smokers. However, certain salivary biomarkers may be potentially useful in combination and alone in the diagnosis of PD, but more systematically robust studies are needed to validate these biomarkers [38]. In our study, higher levels of MDA, SA, salivary cortisol, IL-1β, TIMP, and ICTP were found in smokers compared to non-smokers with PD. In particular, cortisol and IL-1β levels were higher in smokers than in non-smokers. These results were reported by Zhang et al. [43] and are consistent with the report that salivary cortisol levels were significantly higher in smokers with chronic periodontitis than in non-smokers with chronic periodontitis.

However, non-smokers showed high levels of SOD and UA. In addition, although not significant, they had higher levels of activity of LDH and BetaG, MMP-8/TIMP-1 ratio [40], and combined MMP-8 and ICTP. LDH and BetaG activity were significantly decreased in smokers with periodontitis [44], which is similar to the results found in this systematic review, but our results were not statistically significant [5]. IL-1β and MMP-8 were consistent with the diagnostic value of host-derived salivary biomarkers based on the reported sensitivity and specificity in relation to the clinical parameters of the diagnosis of PD in adults [38]. In addition, research results regarding IL-1β are conflicting. Unlike the finding in the current systematic review, another study [45] demonstrated that IL-1β gene expression was lower in smokers with chronic periodontitis than in non-smokers with chronic periodontitis (*p* = 0.003). Currently, there is limited evidence confirming the diagnostic power of salivary biomarkers in the clinical evaluation of PD. Nevertheless, findings from several studies, including this one, are of growing importance for salivary biomarkers and may guide larger and more well-controlled studies of diagnostic accuracy. Although not conclusive, IL-1β is reported to be a promising biomarker for future studies [43].

Saliva is an easy and non-invasive diagnostic fluid that is useful for the diagnosis of early periodontitis, and the possibility of early diagnosis of periodontitis in adolescents, especially boys, based on elevated salivary MMP-8 levels has been reported [46]. Smoking may affect the usefulness of salivary biomarker assays and should always be considered when interpreting biomarker results. Smoking is a risk factor influencing the inflammatory response leading to PD. Therefore, attention should be paid to the disturbance caused by smoking in the interpretation of potential salivary diagnostic test results [47]. While MMP-8 was mainly affected by smoking pack-years, salivary MMP-9 and TIMP-1 are reported to be mainly affected in current smokers or those who have quit smoking within the last 1 year [47]. In addition, a meta-analysis by Lin et al. [48] showed that MMP-8 is currently considered one of the most promising biomarkers for the early diagnosis of periodontitis, but conflicting results were found in several studies, including this study. Overall salivary MMP-8 levels were significantly higher in periodontitis patients compared with healthy controls. However, they reported that higher quality studies are still needed to confirm the conclusions due to the heterogeneity of studies and publication bias [49].

### 4.2. Significance and Limitations of This Review

To the extent we have identified so far, this study is the first systematic review to evaluate salivary biomarkers of PD based on smoking status. To utilize saliva as a reliable diagnostic tool, standardized guidelines for procedures for collecting and processing saliva samples are needed [38]. 

Unfortunately, one limitation of this review is that the systematic literature review process was not performed according to prospero-based registration and protocol. Future research should be conducted according to prospero-based registration for the systematic review protocol. However, we did use the PRISMA 2020 checklist to review the compliance of the systematic literature review studies [29]. 

Differences in study design and methods, index tests, and reference tests have a significant impact on study results [38]. Unfortunately, five studies had a relatively small sample size (*n* < 100) and five had no power calculations [6,8,40,41,42]. In addition, in two studies, the statistical interpretation of the association between PD and salivary biomarkers based on smoking status was limited [5,42]. However, all but one of the seven studies analyzed in our study clearly presented the participant selection and exclusion criteria. 

Future studies should employ larger sample sizes and use validated power calculations to overcome these limitations [50]. In addition, advanced statistical approaches should be used to ensure the validity of the study results [51]. A good approach for participant selection would be to distinguish patients with milder cases of gingivitis and periodontitis. This approach would overcome spectrum bias and prevent overestimation of the results [52]. All seven studies analyzed in our study were observational studies; longitudinal randomized trials may provide more reliable results [50,51]. However, studies of salivary biomarkers useful for diagnosing PD according to smoking status may be difficult to follow-up on at a large scale due to specific problems such as disease progression [53,54] and disease susceptibility in older people [55]. To address the limits and utilize large-scale and cost-effective biological information that would be unsuitable for a smaller sample size, further studies will need to collaborate with major networks such as big data surveyed at the national level.

Regarding saliva sample handling, six studies [5,6,7,8,41,42] collected and stored saliva before clinical assessment. The other study did not mention the process of saliva sampling and storage [40]. In addition, no information about the blinding process was provided, nor were there details about whether information was provided to examiners interpreting the biomarker analysis results. The saliva treatment technique and collection method will affect the change in saliva composition within and between individuals. The composition of saliva can change rapidly with flow rate, type of stimulation, and time. The flow of saliva also drops to nearly zero during sleep; therefore, bedtime and snacking are important variables [34,56].

In general, smoking is known to cause nicotine-induced gingival keratinization and vasoconstriction, thereby masking gingiva bleeding in patients with PD and affecting systemic conditions and various clinical indicators [57]. The fact that systemic conditions would be high-risk confounders for assessing the biomarkers of PD, based on smoking status, should also be noted. In addition, we divided the participants who had already been diagnosed with PD into groups (*n* > 20) based on whether they did or did not smoke. This process caused us to analyze a very limited number of studies. 

Further studies will need to consider the association between smoking and salivary biomarkers as a diagnostic medium of PD by comparing three groups: healthy smokers, smokers with PD, and non-smokers with PD. Analyzing the relevance between smoking habits and salivary biomarkers associated with PD by distinguishing between periodontitis and gingivitis is also necessary. In addition, the standard for determining smoking status, which was the most important variable in this systemic review, differed between studies, and all relied on self-reported responses. Future studies should overcome these problems to improve the quality of studies. In a previous study [56], the change in bleeding on probing over time was significantly higher for female non-smokers with catalase levels of >225 µg/mL than for male non-smokers or male smokers. Thus, analyzing differences by sex is necessary. Comparing and analyzing light smokers and heavy smokers with chronic periodontitis is also a good approach.

## 5. Conclusions

This systematic review summarized evidence regarding effective salivary biomarkers for the diagnosis and monitoring of PD based on smoking status. The levels of cortisol, IL-1β, and MMP-8 were higher in smokers with PD compared to non-smokers. Therefore, in the future, these biomarkers could be used as potential salivary biomarkers for assessing the diagnosis and severity of chronic PD, as well as for helping with the early detection of PD progression. However, in this systematic review, it was confirmed that individual studies had limitations regarding study designs and methods. In future studies, advanced salivary biomarker studies using smoking status should be conducted with well-designed, large-scale, randomized controlled trials.

## Figures and Tables

**Figure 1 ijerph-19-14619-f001:**
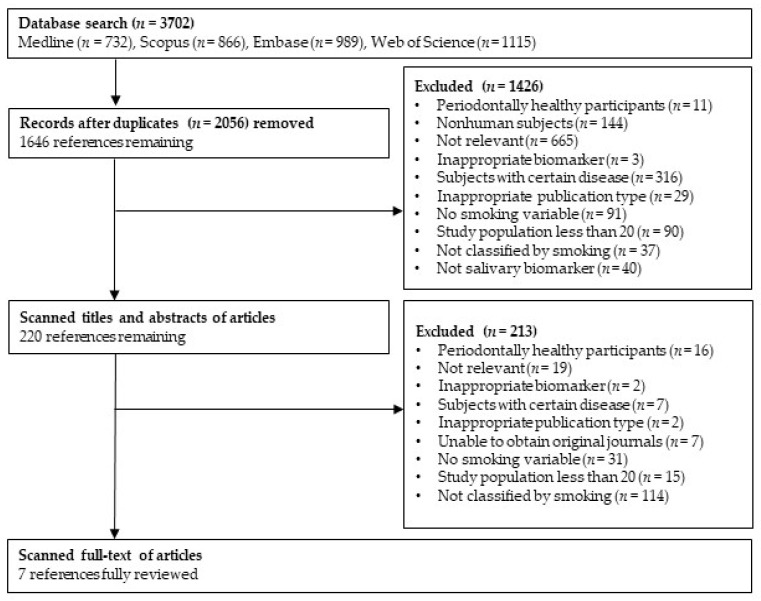
Flow diagram of the literature search strategy.

**Table 1 ijerph-19-14619-t001:** Formulated research questions for the systematic review.

PICOT	Content
Patient	Systemically healthy participants with periodontal disease
Index test	Expression of biomarkers in saliva
Comparison	Clinical parameters (probing pocket depth and clinical attachment loss) or clinical and radiographic parameters (bone loss), irrespective of the diagnostic criteria
Outcome	Differences in salivary biomarker levels based on smoking status
Type of study	Prospective and retrospective study design

**Table 2 ijerph-19-14619-t002:** QUADAS-2 risk of bias assessment.

Authors (Year Published)	Risk of Bias				Applicability Concern		
	Patient Selection	Index Test	Reference Standard	Flow and Timing	Patient Selection	Index Test	Reference Standard
Naresh et al. [6]	☹	?	☺	☺	☺	☺	☺
Ali et al. [5]	☺	?	☺	☺	☺	☺	☺
Bawankar et al. [7]	☹	?	☺	☺	☺	☺	☺
Sharma et al. [38]	☺	?	☺	☺	☺	☺	☺
Gupta et al. [39]	☹	?	☺	☺	☺	☺	☺
Hendek et al. [8]	☹	?	☺	☺	☺	☺	☺
Gursoy et al. [40]	☹	?	☺	☺	☺	☺	☺

☺, low risk; ☹, high risk; ?, unclear risk. QUADAS2, Quality Assessment of Diagnostic Accuracy Studies-2.

**Table 3 ijerph-19-14619-t003:** Descriptive summary of the included studies in the systematic review.

Authors	Country	Study Design	Cases	Age (Year) *	Definition of PD	Definition of Smoking Status	Type of Salivary Biomarkers	Main Conclusion
Naresh et al. [6]	India	Clinicobiochemical study	30 smokers; 30 non-smokers	20–60	Chronic PD: >20 residual teeth, >1 teeth with sites of PPD ≥ 4 mm, and CAL ≥ 4 mm in all four quadrants	Non-smoker: never smoked; smoker: have smoked ≥10 cigarettes/day for ≥5 years	SOD, GPx, MDA, SA	Reduced levels of antioxidant enzymes and elevated levels of lipid peroxidation product could be used as diagnostic markers to measure oxidative stress in PD associated with risk factors such as smoking
Ali et al. [5]	India	Cross-sectional case-control study	50 smokers; 50 non-smokers	30–35	Chronicgeneralized periodontitis: PPD ≥ 5 mm, CAL ≥ 3 mm, and moderate, severe, or generalized disease progression involving >30% of the mouth	Non-smoker: never smoked; smoker: currently smoke ≥5 times/day and have smoked for ≥5 years	Activity of LDH and BetaG	Smoking significantly altered enzymeactivity; however, LDH and BetaG were reliable salivary biomarkers of PD among smokers and non-smokers
Bawankar et al. [7]	India	Observational study	25 smokers; 25 non-smokers	30–65	Untreated moderate to severe CP: PPD ≥ 5 mm and CAL ≥ 5 mm, ≥30% of teeth affected, and radiographic evidence of bone loss	Non-smoker: never smoked; smoker: current smoker and with history of smoking ≥10 cigarettes/day for the last 3 years	Cortisol, IL-1β	Smokers with PD exhibited a significantly higher salivary cortisol and IL-1β; thus, they may have an increased risk of PD and PD severity
Sharma et al. [40]	India	Cross-sectional study	25 smokers; 25 non-smokers	Smokers: 33.32; non-smokers: 34.32	PD: clinically diagnosed with periodontitis in accordance with Russell’s periodontal index score	Non-smoker: no tobacco-related habits; smoker: having tobacco-related habits both smoke and smokeless form	UA, ALB	Saliva can be used as a non-invasive diagnostic fluid with UA and ALB being promising biomarkers in monitoring PD
Gupta et al. [41]	India	Clinicobiochemical study	20 smokers; 20 non-smokers	Smokers: 44.20 ± 7.40; non-smokers: 42.80 ± 8.02	Moderate to severe chronic PD: ≥2 interproximal sites with CAL ≥4 mm or ≥2 interproximal sites with PPD ≥5 mm, not on the same tooth	Non-smoker: never smoked; smoker: smoked ≥1 pack/day for at least past 10 years	MMP-8	MMP-8 is related to periodontium destruction with smoking
Hendek et al. [8]	Turkey	Case-control study	24 smokers; 23 non-smokers	Smokers: 45 (12); non-smokers: 44 (15)	Chronic PD: teeth with 30% periodontal bone loss and ≥2 non-adjacent sites per quadrant with PPD ≥ 5 mm and bleeding on probing	Non-smoker: never smoked; smoker: current smoking of 10 years and ≥10 cigarettes/day	GPx	GPx enzyme activities can be used to determine the protective mechanisms against oxidative stress
Gursoy et al. [42]	Finland	Cross-sectional study	44 smokers; 40 non-smokers ^†^	Smokers: 48.6 ± 5.3; non-smokers: 50.7 ± 4.9	Advanced periodontitis: ≥14 residual teeth with PPD ≥ 4 mm	N/A	MMP-8, MMP-14, TIMP-1, and ICTP	The combinations and ratios of salivary MMP-8, TIMP-1, and ICTP are particularly potential candidates for the detection of advanced periodontitis

* The values are presented as the range, mean ± standard deviation, or median (interquartile range).^†^ The data were calculated by the authors using information available in the article. PD, periodontal disease; PPD, pocket probing depth; CAL, clinical attachment level; SOD, superoxide dismutase; MDA, malondialdehyde; GPx, glutathione peroxidase; LDH, lactate dehydrogenase; BetaG, enzyme activity of β-glucuronidase; UA, uric acid; ALB, albumin; IL-1*β*, salivary interleukin 1-beta; MMP, matrix metalloproteinase; TIMP, tissue inhibitor of matrix metalloproteinase; ICTP, pyridinoline cross-linked carboxyterminal telopeptide of type I collagen; APMA, 4-aminophenylmercuric acetate; ELISA, enzyme-linked immunoassay; EIA, enzyme immunoassay; TR-IFMA, time-resolved immunofluorometric assay; N/A, not available.

**Table 4 ijerph-19-14619-t004:** Salivary biomarker levels based on smoking status.

Authors	Salivary Biomarker	Detection Method	Results *		Significance
			Non-Smoker	Smoker	
Naresh et al. [6]	SOD (U/mL)	Spectrophotometry	50.41 ± 4.25	34.96 ± 4.8	*p* < 0.001
Naresh et al. [6]	MDA (nmol/µL)	Spectrophotometry	0.47 ± 0.11	0.69 ± 0.13	*p* < 0.001
Naresh et al. [6]	Sialic acid (nmol/µL)	Spectrophotometry	0.14 ± 0.02	0.22 ± 0.04	*p* < 0.001
Naresh et al. [6]	GPx (U/L)	Spectrophotometry	124.41 ± 4.74	111.39 ± 6.79	*p* < 0.001
Hendek et al. [8]	GPx (U/µL)	ELISA	30.59 ± 15.06	36.81 ± 9.16	*p* = 0.003
Ali et al. [5]	Activity of LDH (nmol/min/mg)	Spectrophotometry	896.56 ± 264.14	682.58 ± 274.12	N/A
Ali et al. [5]	Activity of BetaG (nmol/min/mg)	Spectrophotometry	76.46 ± 10.43	71.27 ± 12.71	N/A
Bawankar et al. [7]	Cortisol (pg/mL)	ELISA	417.16 ± 99.67	563.40 ± 236.19	*p* < 0.0001
Bawankar et al. [7]	IL-1β (pg/mL)	ELISA	251.35 ± 81.19	278.95 ± 81.40	*p* < 0.0001
Sharma et al. [40]	UA (mg/mL)	Spectrophotometry	1.95 ± 0.423	0.94 ± 0.200	*p* < 0.01
Sharma et al. [40]	ALB (g/mL)	Spectrophotometry	45.44 ± 8.032	47.04 ± 16.032	*p* > 0.05
Gupta et al. [41]	MMP-8 (ng/mL)	ELISA	354.83 ± 29.91	459.16 ± 24.30	*p* < 0.001
Gursoy et al. [42]	MMP-8 (ng/mL)	TR-IFMA	1075.5 (345.2–1715.9)	703.1 (338.6–1646.7)	N/A
Gursoy et al. [42]	MMP-8 (ng/mL)	ELISA	96.7 (61.8–144.7)	83.6 (52.9–114.5)	N/A
Gursoy et al. [42]	TIMP-1 (ng/mL)	ELISA	45.5 (23.3–112.3)	73.0 (42.0–164.0)	N/A
Gursoy et al. [42]	ICTP (ng/mL)	EIA	0.74 (0.56–0.96)	0.75 (0.60–0.94)	N/A
Gursoy et al. [42]	MMP-14 (with APMA) (ng/mL)	ELISA	229.3 (138.1–360.3)	176.2 (112.7–288.3)	N/A
Gursoy et al. [42]	MMP-14 (without APMA) (ng/mL)	ELISA	8.48 (2.21–11.88)	9.31 (4.3–12.8)	N/A
Gursoy et al. [42]	MMP-8/TIMP-1 ratio	TR-IFMA	15.44 ± 21.48	8.26 ± 11.05	N/A
Gursoy et al. [42]	MMP-8/TIMP-1 ratio	ELISA	1.30 ± 1.27	0.69 ± 0.72	N/A
Gursoy et al. [42]	MMP-8 and ICTP combination	TR-IFMA	0.62 ± 0.23	0.58 ± 0.23	N/A
Gursoy et al. [42]	MMP-8 and ICTP combination	ELISA	0.55 ± 0.12	0.52 ± 0.12	N/A

* The values are presented as the mean ± standard deviation or median (interquartile range). SOD, superoxide dismutase; MDA, malondialdehyde; GPx, glutathione peroxidase; LDH, lactate dehydrogenase; BetaG, enzyme activity of β-glucuronidase; UA, uric acid; ALB, albumin; IL-1*β*, salivary interleukin 1-beta; MMP, matrix metalloproteinase; TIMP, tissue inhibitor of matrix metalloproteinase; ICTP, pyridinoline cross-linked carboxyterminal telopeptide of type I collagen; APMA, 4-aminophenylmercuric acetate; ELISA, enzyme-linked immunoassay; EIA, enzyme immunoassay; TR-IFMA, time-resolved immunofluorometric assay; N/A, not available.

## Data Availability

Data sharing is not applicable to this article.

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
