# Peer review of "Evaluation of Salivary Biomarkers of Periodontal Disease Based on Smoking Status: A Systematic Review"

_ijerph, 2022, doi:10.3390/ijerph192114619_

Round 1

Reviewer 1 Report

This thesis is logically described as a systematic literature review. It is believed that it can be used as basic data to predict the risk rate of periodontal disease caused by smoking through the evaluation of Salivary Biomarkers. I look forward to further research.

The main question was to evaluate the evidence, and to highlight the future directions regarding the diagnostic potential of salivary biomarkers associated with PD, based on smoking status. The topic of originality is excellent and relevant in the field. The main question posed the evidence and arguments presented, the references are appropriate, and the tables and figures are also appropriate. Thank you for writing a great paper.

Author Response

We thank you for taking the time and effort necessary to review our manuscript.

Reviewer 2 Report

The authors have given a good try to prepare a systemic review with the title "Evaluation of Salivary Biomarkers of Periodontal Disease, Based on Smoking Status: A Systematic Review "

As the authors mentioning "Differences in study design and methods, index tests and reference tests have a sig- 242 nificant impact on study results. Six studies of this study had a relatively small 243 sample size and no power calculation, and only descriptive statisctics were 244 presented." is the biggest challenge here. 

Introduction:

"It is caused by in- 44 flammation of the surrounding structures of teeth (the gingiva, periodontal ligament and 45 bone), which could lead to tooth loss and contribute to systemic inflammation [2]." 

Comment: if untreated it could lead to tooth loss 

This pharagraph need to be rewritten for better understanding

"At present, periodontitis is usually diagnosed through radiographs and clinical eval- 47 uation including probe pocket depth, probing bleeding, and clinical adhesion level [3]. 48 However, these have relatively limited utility because these findings represent evidence 49 of previous disease activity rather than the present [4]."

The introduction doesn't gives a good flow, strongly recommended to rewrite it!

Materials and methods

It's good to clarify "Conventional clinical parameters" for better understanding

Does this need to be stated under materials and Methods? "The 124 coding was performed by the researchers (HEJ and YL) and four research assistants (HSY, 125 KBK, MHH, YJK). The first author (JWN) reviewed the coding and the categorization of 126 sub-variable groups. The researchers reviewed the coding after it has been coded (HEJ, 127 YL, JHJ). After the review, abnormalities and parts that need confirmation were checked 128 and corrected (HSY, KBK, MHH, YJK). Items that were inconsistent after the review by 129 the researchers were revised and supplemented."

"The seven studies used for the data analysis were coded, based on the year of publi- 120 cation, country, study design, case sample size, age, definition of PD, smoking status, and 121 type of salivary biomarkers for the subgroup analysis on the major categorical variables."

Comment: What was the reason to consider the country?

Results 

Results are not well described in this paper

The main issue is when comparing different markers, the used analysed methods need to get in account as it differs so much as well as the sensitivity of different markers. Couple of study conclusions presented in Table 3 gives more challenges to the authors for a good/ correct comparison. 

Discussion

Some of the facts which need to be stated in the Results are mentioned in the discussion. This need to be considered.

According to my knowledge there are number of studies in the area of "smoking and salivary biomarkers as a diagnostic medium of PD". Check at the PubMed

The Discussion should be rewritten as well after clear Results statements 

Conclusion

I'm very doubtful to this statement: "The results of 307 higher levels of cortisol, IL‐1β, and MMP-8 in smokers with PD, compared to those of 308 nonsmokers, confirmed that salivary biomarkers are useful for diagnosing PD. Hence, 309 salivary biomarkers are useful as an early detection tool for the progression of PD and 310 assessing smoking status and severity of PD in the future." 

References

The References should be updated with recent studies.

Author Response

Thank you for reviewing our research. Below are our responses to your comments and queries. We have incorporates all your recommendations into the revised manuscript. Our revised paper has been checked by a native English speaker (American Journal Experts). 

Our point-by-point response to the reviewer’s comments and suggestions is listed below:

We thank you for taking the time and effort necessary to review our manuscript and provide us with these valuable comments and suggestions. Accordingly, we revised our manuscript and made changes to it. Please note that changes to the manuscript are highlighted in yellow for your convenience.

(Point 1) The authors have given a good try to prepare a systemic review with the title "Evaluation of Salivary Biomarkers of Periodontal Disease, Based on Smoking Status: A Systematic Review "

As the authors mentioning "Differences in study design and methods, index tests and reference tests have a significant impact on study results. Six studies of this study had a relatively small sample size and no power calculation, and only descriptive statisctics were presented." is the biggest challenge here.

(Authors’ Response 1)

Thank you for your constructive comments. Our study aimed to evaluate more meaningful salivary biomarkers based on the results of individual studies. Unfortunately, there were too few analyzable papers that met the PRISMA guidelines, and only the currently presented papers were valid. However, we did our best to achieve the research objectives and tried to reflect your opinions as much as possible.

Introduction:

(Point 2) "It is caused by inflammation of the surrounding structures of teeth (the gingiva, periodontal ligament and bone), which could lead to tooth loss and contribute to systemic inflammation [2]." Comment: if untreated it could lead to tooth loss

This pharagraph need to be rewritten for better understanding

(Authors’ Response 2)

Thank you for the meaningful comments. We have added and revised as per your recommendations:

"It is caused by inflammation of the surrounding structures of teeth, such as the gingiva, periodontal ligaments, and bones; if not treated properly, it could lead to tooth loss and contribute to systemic inflammation [2]."

(Point 3) "At present, periodontitis is usually diagnosed through radiographs and clinical evaluation including probe pocket depth, probing bleeding, and clinical adhesion level [3]. However, these have relatively limited utility because these findings represent evidence of previous disease activity rather than the present [4]."

The introduction doesn't gives a good flow, strongly recommended to rewrite it!

(Authors’ Response 3)

Thank you very much for your professional opinion. We have revised and added the following sentences:

“Since periodontitis often progresses without symptoms, many patients do not receive professional dental care until the periodontal destruction that cannot be treated has progressed [3]. In addition, there is an unmet need for diagnosing PD quickly because PD diagnosis relies on time-consuming clinical measurements [3]. Saliva is an optimal biological fluid to serve as a point-of-care (POC) diagnostic tool for PD. From this point of view, POC diagnosis simplifies diagnosis and improves prognosis, and the feasibility of PD diagnostic testing has been reported [4].

Many promising salivary biomarkers associated with PD have been reported [3].”

Materials and methods

(Point 4) It's good to clarify "Conventional clinical parameters" for better understanding

(Authors’ Response 4)

Thank you for the in-depth comment. "Conventional clinical parameters" have revised and added in detail as follows:

“Clinical parameters (probing pocket depth, clinical attachment loss) or clinical and radiographic parameters (bone loss) irrespective of the diagnostic criteria”

(Point 5) Does this need to be stated under materials and Methods? "The coding was performed by the researchers (HEJ and YL) and four research assistants (HSY, KBK, MHH, YJK). The first author (JWN) reviewed the coding and the categorization of sub-variable groups. The researchers reviewed the coding after it has been coded (HEJ, YL, JHJ). After the review, abnormalities and parts that need confirmation were checked and corrected (HSY, KBK, MHH, YJK). Items that were inconsistent after the review by 129 the researchers were revised and supplemented.“

(Authors’ Response 5)

Thank you for the constructive comments. The roles of the researchers were presented in as much detail as possible, but to reflect your comments, they have been deleted for the sake of brevity.

(Point 6) "The seven studies used for the data analysis were coded, based on the year of publication, country, study design, case sample size, age, definition of PD, smoking status, and type of salivary biomarkers for the subgroup analysis on the major categorical variables.“

Comment: What was the reason to consider the country?

(Authors’ Response 6)

Thank you for the meaningful question. In general, it is considered that there may be systematic differences according to differences in the main race by country and the medical system involved in treatment and examination. Therefore, it was included as a target variable.

Results

(Point 7) Results are not well described in this paper

The main issue is when comparing different markers, the used analysed methods need to get in account as it differs so much as well as the sensitivity of different markers. Couple of study conclusions presented in Table 3 gives more challenges to the authors for a good/ correct comparison.

(Authors’ Response 7)

Thank you for your valuable comments. We have revised and added a careful interpretation of the research results presented in Table 3 as follows:

Table 4 summarizes salivary biomarker levels based on smoking status. Compared to non-smokers, smokers had increased levels of MDA [6], SA [6], salivary cortisol, and IL-1β [7] (p < 0.001). Tissue inhibitor of matrix metalloproteinase (TIMP) [39] and pyridinoline cross-linked carboxyterminal telopeptide of type I collagen (ICTP) [39] were higher in smokers than in non-smokers, but without statistical significance. In addition, albumin [37] was higher in smokers than in non-smokers, but this was not statistically significant (p > 0.05).

On the other hand, the levels of SOD [6] (p < 0.001) and uric acid (UA) [36] (p < 0.01) using spectrophotometry were significantly higher in non-smokers than in smokers. The levels of activity of LDH [5] and BetaG [5], uric acid (UA) [37], MMP-8/TIMP-1 ratio [39], and combination of MMP-8 and ICTP [39] were higher in non-smokers than in smokers, but without statistical significance.

In particular, the salivary biomarkers GPx [6,8] and MMP-8 [37,38] showed conflicting results between smokers and non-smokers when different detection methods were used. In addition, there was a difference in descriptive statistics in different studies on MMP-8 using enzyme-linked immunosorbent assay (ELISA), where the level of MMP-14 was higher with APMA than without APMA.

Discussion

(Point 8) Some of the facts which need to be stated in the Results are mentioned in the discussion. This need to be considered. According to my knowledge there are number of studies in the area of "smoking and salivary biomarkers as a diagnostic medium of PD". Check at the PubMed.

The Discussion should be rewritten as well after clear Results statements

(Authors’ Response 8)

Thank you very much for your thoughtful comment. In response to your comments, we have significantly revised the discussion section as follows:

4.1. Evidence for Salivary Biomarkers, Based on Smoking Status

  • As a result of a systematic literature review on salivary biomarkers for PD diagnosis by smoking status, some markers showed significant differences between smokers and non-smokers. However, certain salivary biomarkers may be potentially useful in combination and alone in the diagnosis of PD, but more systematically robust studies are needed to validate these biomarkers [35]. In our study, higher levels of MDA, SA, salivary cortisol, IL-1β , TIMP, and ICTP were found in smokers compared to non-smokers with PD. In particular, cortisol and IL-1β levels were higher in smokers than in non-smokers. These results were reported by Zhang et al. [41] and are consistent with the report that salivary cortisol levels were significantly higher in smokers with chronic periodontitis than in non-smokers without chronic periodontitis.
  • On the other hand, non-smokers showed high levels of SOD and UA. In addition, although not significant, they had higher activity of LDH and BetaG, MMP-8/TIMP-1 ratio [39], and combined MMP-8 and ICTP. LDH and BetaG activity were previously reported to be significantly decreased in smokers with periodontitis [42], which is similar to the results found in this systematic review, but our results were not significant [5]. IL-1β and MMP-8 were consistent with the diagnostic value of host-derived salivary biomarkers based on the reported sensitivity and specificity in relation to the clinical parameters of the diagnosis of PD in adults [35]. In addition, research results regarding IL-1β are conflicting. Unlike the findings in the current systematic review study, another study [43], demonstrated that the IL-1β gene expression was lower in smokers with chronic periodontitis than in non-smokers with chronic periodontitis (p = 0.003).
  • Currently, there is limited evidence confirming the diagnostic power of salivary biomarkers in the clinical evaluation of PD. Nevertheless, findings from several studies, including this one, are of growing importance for salivary biomarkers and may guide larger and more well-controlled studies of diagnostic accuracy. Although not conclusive, IL-1β is reported to be a promising biomarker for future studies [41].
  • Saliva is an easy and non-invasive diagnostic fluid that is useful for the diagnosis of early periodontitis, and the possibility of early diagnosis of periodontitis in adolescents, especially boys, based on elevated salivary MMP-8 levels has been reported [44]. Smoking may affect the usefulness of salivary biomarker assays and should always be considered when interpreting biomarker results. Smoking is a risk factor influencing the inflammatory response leading to PD. Therefore, attention should be paid to the disturbance caused by smoking in the interpretation of potential salivary diagnostic test results [45]. While MMP-8 was mainly affected by smoking pack-years, salivary MMP-9 and TIMP-1 are reported to be mainly affected by current smokers or those who have quit smoking within the last 1 year [45]. In addition, a meta-analysis by Lin et al [46] showed that MMP-8 is currently considered as one of the most promising biomarkers for the early diagnosis of periodontitis, but conflicting results were found in several studies including this study. Overall, salivary MMP-8 levels were significantly higher in periodontitis patients compared with healthy controls. However, they reported that higher quality studies are still needed to confirm the conclusions due to the heterogeneity of studies and publication bias [47].

4.2. Significance and Limitations of This Review

  • Unfortunately, five studies in this review had relatively small sample sizes (n < 100) and five had no power calculations [6-8,36,37]. Also, in two studies, statistical interpretation of the association between PD and salivary biomarkers based on smoking status was limited [5,39]. However, all but one of the seven studies analyzed in our study clearly presented the participant selection and exclusion criteria.

Conclusion

(Point 10) I'm very doubtful to this statement: "The results of higher levels of cortisol, IL‐1β, and MMP-8 in smokers with PD, compared to those of nonsmokers, confirmed that salivary biomarkers are useful for diagnosing PD. Hence, salivary biomarkers are useful as an early detection tool for the progression of PD and assessing smoking status and severity of PD in the future."

(Authors’ Response 10)

Thank you for your thoughtful comments. We have revised the sentence as follows:

  • The level of cortisol, IL-1β, and MMP-8 levels were higher in smokers with PD compared to non-smokers. Therefore, in the future, these biomarkers could be used as potential salivary biomarkers for assessing the diagnosis and severity of chronic PD as well as helping in early detection of PD progression”.

(Point 11) The References should be updated with recent studies.

(Authors’ Response 11)

In response to your constructive comments, we have updated some references with recent studies in red. Thank you very much.

We have incorporated all your recommendations and corrections in the revised manuscript. Please let us know if you want us to make any further modifications and we shall be glad to do so. Thank you very much.